# Capability Traps in DPO

**Marco Pollanen** [1]

## Abstract

Direct Preference Optimization (DPO) is often tuned by selecting checkpoints with high preference margins. We show that this selection rule can fail. Across dense $\beta$ sweeps in three 7B open-weight families under controlled DPO recipes, the DPO preference margin can strongly anticorrelate with capability probes, with the strongest observed case in LLaMA-2-7B logic probes (Pearson $r = -0.91$, $n = 13$). In Mistral-7B, matched-duration controls show that transient exposure to higher $\beta$ produces persistent arithmetic ($d_z = 3.73$, $p = 0.002$) and format ($d_z = 3.38$, $p = 0.003$) degradation relative to a duration-matched constant-$\beta$ run, while logic and sycophancy show no significant effect. An expanded 14-probe logic suite further shows that aggregate capability scores can hide stable opposition between probe clusters: direct-inference probes are consistently negative while fallacy-detection probes are consistently positive, with 13 of 14 probes sign-stable across 3 seeds and 3 $\beta$ values. Sensitivity profiles differ across architectures: Mistral-7B shows elevated seed variance near $\beta \approx 10^{-2}$, LLaMA-2-7B shows capability-specific rigidity, and Qwen-1.5-7B trades off smoothly. These findings motivate probe-resolved $\beta$ sweeps rather than margin-based checkpoint selection.

## 1. Introduction

The standard DPO pipeline sweeps the regularization parameter $\beta$, selects the checkpoint with the best preference margin, and validates on aggregate benchmarks. We document three empirical failure modes, observed across controlled experiments on three 7B architectures.

[1] Department of Mathematics, Trent University, Peterborough, ON, Canada. Correspondence to: Marco Pollanen <marcopollanen@trentu.ca>.

*Proceedings of the 43rd International Conference on Machine Learning*, Seoul, South Korea. PMLR 306, 2026. Copyright 2026 by the author(s).

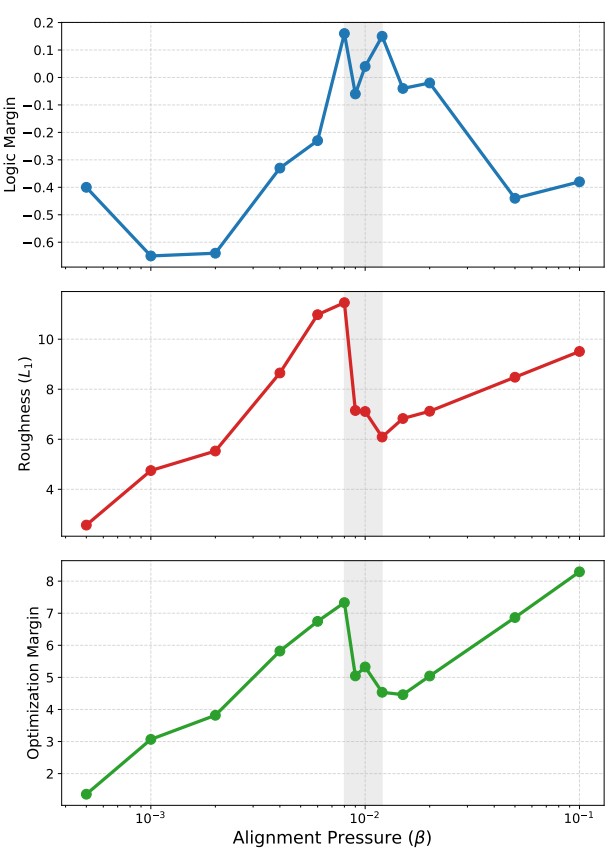

*Figure 1.* **Three ways standard DPO tuning can fail.** Sweeping alignment pressure $\beta$ in Mistral-7B under our DPO recipe. **Top:** The 3-probe logic aggregate (canonical seed) crosses zero at isolated $\beta$ points. With an expanded 14-probe suite, the aggregate is positive at all $\beta$ tested; the apparent crossing reflects stable opposition between probe clusters (§3.5), not reasoning emergence. **Middle:** Training roughness varies sharply with $\beta$. It is useful as a screening signal but is not itself a capability diagnostic. **Bottom:** The DPO preference margin grows roughly monotonically with $\beta$, but margin and logic-probe capability can anticorrelate ($r = -0.91$ for LLaMA-2-7B; §3.3), so margin-based selection can prefer capability-impaired models.

1. **Proxy failure.** The DPO preference margin can anticorrelate with logic-probe capability ($r = -0.91$ for LLaMA-2-7B logic), so the optimized proxy and the measured target can point in opposite directions.

2. **Capability-specific path dependence.** Under

matched-duration controls, exposure to higher $\beta$ during training induces persistent degradation in arithmetic and format probes ($p < 0.005$, paired $t$-test, $n = 5$), but not in logic or sycophancy. Annealing through a high-$\beta$ region is not equivalent to training at the lower $\beta$ alone for these capabilities.

3. **Architecture-dependent sensitivity.** The same protocol produces qualitatively different regimes across architectures: elevated seed variance and 3-probe-aggregate sign changes in Mistral-7B near $\beta \approx 10^{-2}$, capability-specific rigidity in LLaMA-2-7B, and smooth trade-offs in Qwen-1.5-7B.

**Relationship to the submitted version.** Additional controls run during the review period refine two claims from our submitted draft (titled "The Viscosity of Logic"). First, a matched-duration experiment shows that path dependence in Mistral-7B is strongest for arithmetic and format, not logic, correcting an A-versus-B comparison that confounded $\beta$-history with training duration (§3.4). Second, an expanded 14-probe logic suite shows that the originally reported near-zero "logic-positive pocket" aggregate reflected stable opposition between probe clusters rather than a uniform reasoning transition (§3.5, Appendix A). We center the paper on the two findings best supported by the revised experiments: proxy–capability decoupling and capability-specific path dependence.

**Contributions.** We present the controlled $\beta$-sweep methodology and resulting capability landscape (§3.1), the architecture comparison and seed-variance scan (§3.2), the margin–capability anticorrelation (§3.3), the matched-duration path-dependence test (§3.4), and the expanded probe suite with its probe-cluster structure (§3.5). Auxiliary findings on structural collapse and a cross-architecture sensitivity scan are in Appendices C and D. We close with a practical checklist for alignment hyperparameter sweeps (Table 8).

## 2. Background and Setup

### 2.1. Direct Preference Optimization

DPO (Rafailov et al., 2023) optimizes pairwise preferences with a Kullback–Leibler divergence term relative to a reference policy and a temperature-like scalar $\beta$. In the standard objective, $\beta$ scales the preference term and, in common implementations, is inversely related to the effective strength of staying close to the reference policy. We accordingly treat larger $\beta$ as stronger preference-optimization pressure and refer to it as "higher alignment pressure" in this paper.

We use two distinct margin notions throughout. (i) The *DPO preference margin* is the optimized proxy itself, denoted $m_t$

over training steps and aggregated at the end of training. (ii) *Probe margins* are length-normalized log-probability differences on fixed capability probes, where positive values indicate that the correct continuation is preferred to the distractor under the probe scoring rule. Standard practice tunes $\beta$ for downstream performance (Tunstall et al., 2023); we instead sweep it systematically to map the capability landscape.

**Ruling out metric artifacts.** Apparent emergence can arise from nonlinear evaluation metrics (Schaeffer et al., 2023). Our key discontinuity evidence is *path dependence*: two training paths that end at identical $\beta$ are evaluated identically yet yield measurably different probe margins, implying different final parameter states under our protocol. While DPO variants can exhibit length effects (Park et al., 2024; Xu et al., 2024), our main qualitative results concern sign changes and path dependence in fixed probes rather than aggregate generation length.

### 2.2. Experimental Protocol

**Frozen configuration principle.** Each $\beta$ point begins from a fresh base model copy with an identical recipe, isolating $\beta$ effects from confounds. Deviations (e.g., path-dependence tests) are deliberate and labeled.

**Models.** We selected three 7B families to span the open-weight ecosystem rather than to cherry-pick favorable results: Mistral-7B (`mistralai/Mistral-7B-v0.1`), LLaMA-2-7B (`NousResearch/Llama-2-7b-hf`), and Qwen-1.5-7B (`Qwen/Qwen1.5-7B`).

**Hyperparameters and configuration disclosure.** Our experiments use two LoRA configurations, depending on training script lineage; we state both explicitly so that the relevant numbers are unambiguous. Unless otherwise noted, all runs use learning rate $5 \times 10^{-5}$, AdamW, batch size 1 with gradient-accumulation 8 (effective batch 8), and LoRA dropout 0.05.

- **Canonical sweep, multi-seed variance, architecture comparison, learning-rate stress test, and expanded probe evaluations:** 200 optimizer steps, LoRA rank $r = 16$, $\alpha = 32$.

- **Path-dependence experiments (original Path A/B and the new Path C matched-duration control) and the 1000-step extended sweep:** LoRA rank $r = 8$, $\alpha = 16$. Path A: 200 steps at terminal $\beta$. Path B: 200 steps at $\beta = 0.02$ followed by 200 steps at terminal $\beta$ (400 total). Path C: 400 steps at terminal $\beta$ (matched duration to B).

We tabulate findings per configuration so the reader can

verify what was observed under each setting; the qualitative direction of effects (proxy decoupling, the direction of path dependence, the probe-cluster sign structure) reproduces in both, but precise effect sizes are not directly comparable across configurations.

**Beta selection.** We employed a logarithmic sweep

$$\beta \in \left\{ \begin{array}{l} 0.0005, 0.001, 0.002, 0.004, 0.006, 0.008, 0.009, \\ 0.010, 0.012, 0.015, 0.020, 0.050, 0.100 \end{array} \right\},$$

with increased resolution near $\beta \approx 0.006$–$0.015$, where preliminary coarse sweeps indicated heightened sensitivity. The 1000-step extended sweep uses a 7-point subset; the expanded-probe single-seed sweep uses a 6-point subset; details in the relevant tables.

**Canonical capability probes.** Logic: syllogistic reasoning, ordering (3 probes). Arithmetic: multi-digit operations (3 probes). Format: JSON generation, boolean constraints (4 probes). Sycophancy: resistance to false claims (2 probes), following Perez et al. (2022). Negation: negated-query handling (2 probes). Probe margins are length-normalized log-probability differences; positive margins indicate that the correct continuation is preferred. Full probe text for the canonical and expanded suites is in Appendix A.

**Expanded logic probes.** To address the thinness of the original 3-probe logic suite, we evaluate an expanded 14-probe logic suite spanning three difficulty levels (4 L1 / 6 L2 / 4 L3 probes), plus additional probes in other categories (27 probes total). All probes are released with the reproducibility archive.

**Training roughness.** We quantify optimization turbulence by *roughness*, defined as the mean absolute stepwise change in the preference-margin trajectory: $R = \frac{1}{T-1} \sum_{t=2}^{T} |m_t - m_{t-1}|$. We use roughness as a screening signal for sensitive $\beta$ regions, not as a capability diagnostic.

## 3. Results

### 3.1. Non-Monotonic Capability Dynamics in Mistral-7B

At fixed 7B scale, varying $\beta$ induces non-monotonic capability changes in Mistral-7B. Table 1 presents the canonical 13-point sweep at 200 steps.

**The near-zero aggregate is sign opposition, not a transition.** At $\beta \in \{0.008, 0.010, 0.012\}$ the 3-probe logic aggregate is barely positive. Probe decomposition (§3.5) shows this is driven by syllogism_2 (+2.38 at $\beta = 0.01$) crossing in opposition with consistently negative syllogism_1 and ordering. With the expanded 14-probe suite, the logic aggregate is positive at every $\beta$ tested.

*Table 1.* Mistral-7B canonical sweep at 200 steps (single seed). Bold: points where the aggregated 3-probe logic margin is positive. As discussed in §3.5, this near-zero aggregate reflects probe-cluster sign opposition rather than a uniform reasoning transition. **Margin** denotes the final DPO preference margin (optimized proxy), not a probe score.

| $\beta$ | Logic | Arith | Format | Syco | Neg | Margin |
|---|---|---|---|---|---|---|
| 0.0005 | −0.40 | +2.81 | +4.44 | +1.09 | +4.25 | 1.4 |
| 0.001 | −0.65 | +2.34 | −0.62 | −1.81 | +3.59 | 3.1 |
| 0.002 | −0.64 | +2.75 | −0.13 | +1.56 | +5.12 | 3.8 |
| 0.004 | −0.33 | +2.68 | −0.52 | +0.82 | +3.88 | 5.8 |
| 0.006 | −0.23 | +2.66 | −1.11 | −0.70 | +3.75 | 6.7 |
| **0.008** | **+0.16** | +2.67 | −1.11 | +1.29 | +3.61 | 7.3 |
| 0.009 | −0.06 | +2.71 | −1.00 | +1.55 | +3.83 | 5.0 |
| **0.010** | **+0.04** | +2.75 | −0.90 | +1.25 | +3.66 | 5.3 |
| **0.012** | **+0.15** | +2.72 | −1.09 | +0.87 | +3.67 | 4.5 |
| 0.015 | −0.04 | +2.68 | −1.14 | +1.30 | +3.48 | 4.5 |
| 0.020 | −0.02 | +2.67 | −1.17 | +0.39 | +3.17 | 5.0 |
| 0.050 | −0.44 | +2.65 | −1.24 | +0.40 | +2.47 | 6.9 |
| 0.100 | −0.38 | +2.47 | −1.24 | −0.09 | +2.02 | 8.3 |

The bolded crossings in Table 1 are aggregate artifacts, not reasoning emergence.

**Multi-seed variance localizes a sensitive region.** Across 5 seeds, the 3-probe logic variance peaks near $\beta = 0.006$, and the fraction of seeds with positive aggregate logic is $3/5$ at $\beta = 0.006$, $4/5$ at $\beta = 0.008$, $4/5$ at $\beta = 0.012$, and $0/5$ at $\beta \geq 0.015$. We refer to this as a *seed-sensitive region* rather than a uniformly ordered pocket; identical hyperparameters can produce opposite aggregate signs across seeds.

**1000-step validation.** To address whether the 200-step picture reflects an under-trained regime, we ran a 7-point Mistral-7B sweep to 1000 steps. Non-monotonicity persists at $5\times$ longer training; training curves show late-stage plateaus and small gradient norms, weakening an under-training interpretation. The 1000-step sweep also yields the cleanest single-point illustration of proxy–capability dissociation in our data: at $\beta = 0.010$, the preference margin spikes to 41.1 (the maximum across the sweep) while aggregate logic reaches its minimum of $-0.96$. Full per-point results, including the syllogism-probe sign reversal at $\beta = 0.010$, are in Appendix E.

**Hierarchy of response.** Different capabilities respond at different pressures. Format, sycophancy, and negation show positive margins at the lowest $\beta$ tested (0.0005); logic requires $\beta \approx 0.008$ for the 3-probe aggregate to cross zero, an order of magnitude higher. This is consistent with a hierarchy in which surface compliance is more readily achieved than the kind of probe-level changes that drive the logic aggregate.

*Table 2.* Three architectures at $\beta = 0.01$ (200 steps, seed 1).

|        | Mistral   | LLaMA     | Qwen    |
| ------ | --------- | --------- | ------- |
| Logic  | +0.04     | −0.95     | +1.08   |
| Arith  | +2.75     | +1.62     | +3.00   |
| Format | −0.90     | −1.98     | −0.70   |
| Syco   | +1.25     | +1.71     | +3.01   |
| Neg    | +3.66     | +1.58     | +0.42   |
| Response | Plastic | Selective | Smooth  |

*Table 3.* Variance at $\beta = 0.006$ (5 seeds). Mistral-7B shows 27–2259× higher variance than LLaMA-2-7B across the listed capabilities.

| Capability | Mistral $\sigma^2$ | LLaMA $\sigma^2$ | Ratio  |
| ---------- | ------------------ | ---------------- | ------ |
| Logic      | 0.016              | 0.0006           | 27×    |
| Format     | 0.015              | 0.00001          | 1751×  |
| Syco       | 0.783              | 0.0003           | 2259×  |

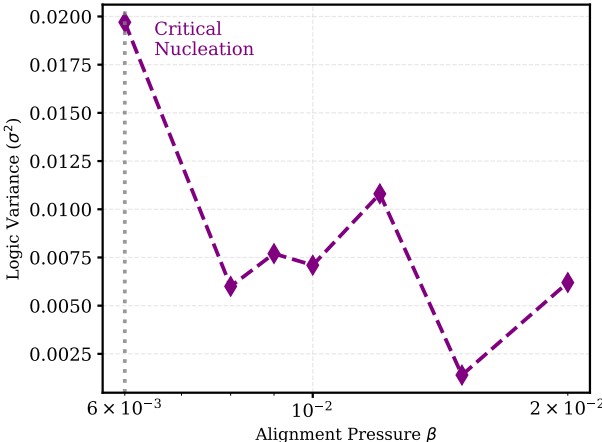

*Figure 2.* Logic variance across 5 seeds peaks at $\beta = 0.006$, marking the region where 3-probe aggregate outcomes are maximally seed-sensitive in Mistral-7B.

## 3.2. Architecture-Dependent Responses

The same protocol yields qualitatively different regimes across architectures (Table 2).

We use these labels descriptively: *plastic* denotes qualitative reorganization with elevated seed sensitivity, *selective* denotes capability-specific rigidity, and *smooth* denotes continuous trade-offs without sharp discontinuities under our probe set.

To quantify sensitivity near the sensitive region, we measure seed variance at $\beta = 0.006$ across 5 seeds (Table 3).

Mistral-7B exhibits 27–2259× higher variance than LLaMA-2-7B across logic, format, and sycophancy probes at $\beta = 0.006$ (Figure 2, Table 3). However, LLaMA-2-7B's rigidity is capability-specific: while its logic remains essentially frozen across the sweep, sycophancy and format margins still vary with $\beta$, indicating compartmentalized response modes.

**Learning-rate stress test (LLaMA-2-7B).** To test whether LLaMA-2-7B's flat logic profile reflects insufficient step size, we varied the learning rate by 20× (1e-5, 5e-5, 2e-4) at $\beta \in \{0.006, 0.01, 0.02\}$. Logic remained negative across all 9 conditions with total spread 0.125, indicating the rigidity is architectural rather than a step-size effect. Per-condition values are in Appendix F.

## 3.3. Margin–Capability Decoupling

A key practical risk is that the optimized DPO proxy can decouple from capability. Models can appear more aligned by the optimized metric while becoming less capable on reasoning tasks (Table 4).

For LLaMA-2-7B logic, Pearson $r = -0.91$ (two-sided

*Table 4.* Margin–capability correlation across the Mistral-7B and LLaMA-2-7B canonical sweeps ($n = 13$ grid points). For LLaMA-2-7B logic, $r = -0.91$ (two-sided $p < 10^{-4}$).

| Capability | Mistral | LLaMA |
| ---------- | ------- | ----- |
| Logic      | +0.27   | −0.91 |
| Format     | −0.70   | —     |
| Neg        | −0.64   | —     |

**Note:** — indicates correlation not reported because variance was insufficient to support a meaningful estimate under our protocol.

$p < 10^{-4}$, $n = 13$), indicating that higher DPO preference margin is associated with lower logic-probe capability under our sweep. This is consistent with an internal instance of Goodhart's law: increasing alignment pressure can improve the optimized proxy (margin) while degrading the measured target capability (logic). This connects to reward over-optimization (Gao et al., 2023), but manifests here as capability degradation rather than reward hacking.

Benchmark gains can coincide with internal probe degradation (Figure 3). In our Mistral-7B sweep, GSM8K accuracy anticorrelates with the aggregated logic-probe margin (Pearson $r = -0.89$, $p = 0.017$, $n = 6$ valid points after excluding the degenerate $\beta = 0.002$ score), peaking at $\beta = 0.02$ while logic margins are strongly negative. Aggregate behavioral benchmarks can therefore mask internal probe degradation under alignment pressure.

The 1000-step extended sweep (Appendix E) shows this dissociation survives longer training: the preference margin at $\beta = 0.010$ rises to 41.1 while aggregate logic is −0.96. The dissociation is not an artifact of incomplete optimization.

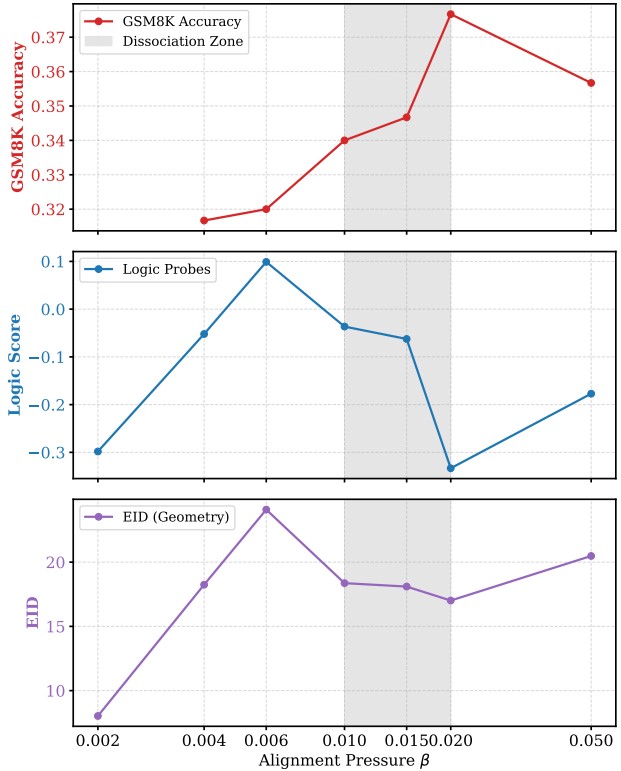

*Figure 3.* **Capability–benchmark dissociation in Mistral-7B.** GSM8K accuracy rises with $\beta$ while logic-probe margins decline. The Effective Intrinsic Dimensionality (EID) of the weight space peaks at $\beta = 0.006$ and decreases at higher $\beta$. Shaded region: the dissociation zone where benchmarks improve despite internal probe degradation. We do not interpret EID mechanistically but report it as an additional geometric signal co-localized with the sensitive region. For GSM8K we treat $\beta = 0.002$ as missing data due to a degenerate zero score in our pipeline.

### 3.4. Capability-Specific Path Dependence

Training path affects final capability even at identical terminal $\beta$, but the effect is capability-specific. We report a matched-duration control to isolate path effects from training duration.

We compare three paths to $\beta = 0.01$:

- **Path A (200 steps, constant).** $\beta = 0.01$ for 200 steps.

- **Path B (anneal, 400 steps).** $\beta = 0.02$ for 200 steps, then $\beta = 0.01$ for 200 steps.

- **Path C (constant, 400 steps).** $\beta = 0.01$ for 400 steps. Matched-duration control for B.

An A–B comparison alone confounds $\beta$-history with training duration (200 vs. 400 steps). The correct test for $\beta$-history effects is C versus B: same total optimization budget, different exposure to $\beta = 0.02$. We ran A, B, and C with $n = 5$ seeds on identical hardware (Table 5).

*Table 5.* Three-path comparison ($n = 5$ seeds, $r = 8$, $\alpha = 16$). Path scores at terminal $\beta = 0.01$. C vs. B statistics (paired $t$-test) test capability-specific path dependence under matched duration. "Consistent" counts seeds with $\text{sign}(C - B) > 0$.

| Capability | Path A | Path B | Path C | $C - B$ | $p$ | $d_z$ |
|---|---|---|---|---|---|---|
| Arith | +2.75 | ∼+0.8 | ∼+2.7 | **+1.87** | **0.002** | 3.73 |
| Format | −0.90 | ∼−1.6 | ∼−0.6 | **+1.08** | **0.003** | 3.38 |
| Neg | +3.66 | ∼+2.1 | ∼+3.4 | +1.30 | 0.142 | 0.91 |
| Syco | +1.25 | ∼−1.7 | ∼−0.3 | +1.40 | 0.311 | 0.58 |
| Logic | +0.04 | ∼−0.1 | ∼−0.2 | −0.12 | 0.663 | −0.24 |

**Note:** Path B and Path C scores are approximate seed means; per-seed values used for the paired test. Consistent direction across 5/5 seeds for Arith, Format, and Neg; mixed for Syco and Logic.

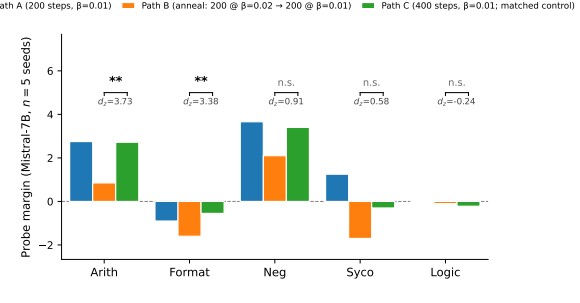

*Figure 4.* **Capability-specific path dependence.** Three-path comparison ($n = 5$ seeds): A (200 steps, constant), B (anneal, 400 steps), C (400 steps, constant; matched-duration control for B). C minus B isolates the effect of transient high-$\beta$ exposure from training duration. Arithmetic and format show significant degradation under B relative to C; logic does not.

The matched-duration test supports *capability-specific* path dependence in Mistral-7B: relative to a 400-step constant-$\beta$ run, transient exposure to higher $\beta$ significantly degrades arithmetic ($d_z = 3.73$, $p = 0.002$, 5/5 seeds consistent) and format ($d_z = 3.38$, $p = 0.003$, 5/5 seeds consistent). Logic and sycophancy do not reach significance ($p > 0.3$).

The implication is that annealing schedules should not be assumed safe for arithmetic and format capabilities under DPO: in our protocol, high-$\beta$ exposure induces degradation that does not recover when $\beta$ is subsequently reduced. This persistent capability loss aligns with recent findings on the *loss of plasticity* in continual learning (Dohare et al., 2024), suggesting DPO can induce similar rigidity in weight space for some capabilities but not others.

### 3.5. Probe Structure: Capabilities Are Bundles

Aggregate metrics obscure probe-level heterogeneity. At $\beta = 0.01$, the canonical 3-probe Mistral-7B logic aggregate is $+0.04$ (barely positive) but compresses dramatically different behaviors (Table 6).

Within logic: `syllogism_2` is always positive (robust);

*Table 6.* Canonical-probe decomposition at $\beta = 0.01$. Logic aggregate (+0.04) hides 2/3 negative probes.

| Category | Probe | Score | Sign |
|---|---|---|---|
| Logic | syllogism_1 | −0.62 | − |
| | syllogism_2 | +2.38 | + |
| | ordering | −1.62 | − |
| Format | json_simple | +0.35 | + |
| | json_key | −0.04 | − |
| | strict_bool | −4.48 | − |
| | strict_json | +0.56 | + |

*Table 7.* Mistral-7B expanded logic suite (single seed). L1: simple direct inference (4 probes). L2: moderate reasoning, including fallacy detection (6 probes). L3: multi-step / harder (4 probes). The full-suite aggregate is positive at every $\beta$, in contrast to the canonical 3-probe aggregate.

| $\beta$ | L1 | L2 | L3 | Aggregate |
|---|---|---|---|---|
| base | +0.27 | +0.34 | −0.19 | +0.17 |
| 0.001 | −1.88 | +4.31 | +1.38 | +1.71 |
| 0.006 | −1.89 | +3.62 | +1.16 | +1.34 |
| 0.010 | −1.43 | +3.14 | +0.55 | +1.09 |
| 0.020 | −1.88 | +3.42 | +1.22 | +1.28 |
| 0.050 | −1.38 | +4.03 | +0.77 | +1.56 |

syllogism_1 is positive only at $\beta = 0.001$ (fragile); ordering is never positive (frozen). A model "passing" the aggregate may fail 2/3 of constituent tasks. Within the logic bundle itself, probes trade off: syllogism_1 and syllogism_2 show $r = -0.71$ across the $\beta$ sweep ($n = 13$, $p = 0.006$).

**Expanded 14-probe logic suite.** To address whether the canonical 3-probe set is too thin, we evaluate an expanded 14-probe logic suite spanning three difficulty levels (Table 7; full probe text in Appendix A). The aggregate logic margin under this suite is positive at every $\beta$ tested in Mistral-7B, including the points where the 3-probe aggregate is negative.

A striking pattern emerges: L1 (direct inference) probes become consistently more negative after DPO, while L2 probes (which include fallacy-detection items requiring the model to reject invalid syllogisms) become consistently more positive. The pattern is not strictly a function of difficulty label, however; the cleanest split is between *direct-inference probes* (consistently negative after DPO) and *fallacy-detection probes* (consistently positive after DPO), and this split crosses difficulty levels. For instance, simple_negation is labeled L1 but is always positive, while ordering is labeled L2 but is always negative.

**Multi-seed sign stability.** We ran the expanded suite with 3 seeds at $\beta \in \{0.006, 0.01, 0.02\}$ in Mistral-7B. Out of 14 logic probes, 13 produce identical signs across all 3 seeds

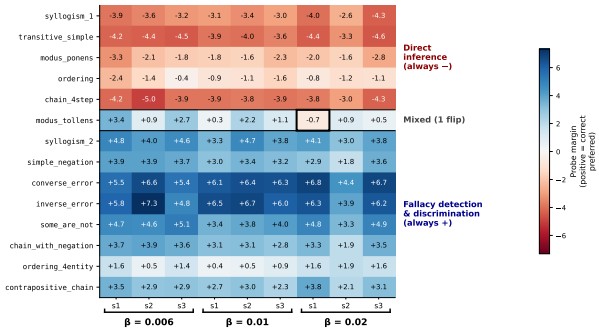

*13 of 14 probes are sign-stable across 3 seeds × 3 β values; the single sign flip (boxed) is in modus_tollens at β=0.02, seed 1.*

*Figure 5.* **Probe-cluster sign stability under DPO.** Sign of the probe margin for each of 14 logic probes across 3 random seeds and 3 $\beta$ values (Mistral-7B, expanded suite). 13/14 probes are sign-stable across all 9 conditions. Direct-inference probes (top cluster) are consistently negative after DPO; fallacy-detection and discrimination probes (bottom cluster) are consistently positive. The 3-probe canonical aggregate crosses zero because it mixes probes from both clusters (syllogism_2 from the positive cluster with syllogism_1 and ordering from the negative cluster).

at all 3 $\beta$ values; only modus_tollens flips once (at $\beta = 0.02$, seed 1). The five always-negative probes are syllogism_1, ordering, transitive_simple, modus_ponens, and chain_4step; all require direct deduction from premises to a conclusion. The eight always-positive probes are syllogism_2, simple_negation, converse_error, inverse_error, some_are_not, chain_with_negation, ordering_4entity, and contrapositive_chain; most require recognizing an invalid inference or otherwise discriminating among options rather than synthesizing a conclusion. Full per-probe results are in Appendix A.

**Interpreting narrow positive bands in the 3-probe aggregate.** The 3-probe aggregate's near-zero crossings at narrow $\beta$ values are a probe-aggregation effect, not a reasoning transition. With more probes, the aggregate is uniformly positive; the sign of each individual probe is largely $\beta$-invariant. The systematic anticorrelation between probe clusters is the real signal.

**Bundle structure beyond logic.** The bundle structure extends beyond logic: sycophancy probes flat_earth and bad_math are anti-correlated across the $\beta$ sweep in Mistral-7B ($r = -0.50$). Within a named capability, constituent tasks can trade off under the same alignment pressure. Format probes show a related pattern: Mistral-7B starts with high format integrity that erodes with $\beta$, while LLaMA-2-7B starts negative and never recovers; this aligns with our path-dependence result for format and is detailed in Appendix C. An auxiliary associative-logic scan run under a faster training schedule is reported in Appendix D.

# 4. Discussion

## 4.1. Scope and Boundaries

We claim only what our controlled experiments support: at 7B scale and under our DPO recipes, varying $\beta$ can produce non-monotonic probe-margin changes, seed-sensitive boundaries near $\beta \approx 10^{-2}$, and capability-specific path dependence at identical terminal $\beta$. We do not claim universal critical exponents, mechanistic explanations, or that the sensitive region's location generalizes to other scales. Our cross-architecture comparisons do not imply that LLaMA-2-7B or Qwen-1.5-7B are globally unresponsive to alignment; both show changes in non-logic probe categories. The architecture-, capability-, and probe-cluster-specific response modes we observe warrant targeted follow-up.

## 4.2. Three Response Modes

Based on the observed variance and stability profiles, we classify the three architectures into descriptive response modes.

**Plastic (Mistral-7B).** Elevated seed variance and 3-probe-aggregate sign changes near $\beta \approx 10^{-2}$. Under matched-duration controls, arithmetic and format exhibit persistent path dependence.

**Selective (LLaMA-2-7B).** Capability-specific rigidity: logic remains essentially frozen across the $\beta$ sweep, and is robust even to a $20\times$ learning-rate range (spread 0.125 across 9 conditions). Sycophancy and format probes still vary with $\beta$. The DPO preference margin and logic-probe capability are most strongly anticorrelated in this architecture ($r = -0.91$).

**Smooth (Qwen-1.5-7B).** Capabilities trade off gradually across the $\beta$ sweep without sharp discontinuities in our probe margins.

This framework explains why a single "best $\beta$" does not exist: the optimal pressure depends on architecture, capability, and the probe used to measure capability.

## 4.3. Mechanistic Hypotheses

We provide empirical constraints, not internal explanations. We list testable hypotheses for follow-up.

**Probe-cluster asymmetry.** DPO trains a discrimination task (prefer chosen over rejected). The probes that improve under DPO in our data (fallacy detection, invalid-syllogism rejection, discrimination among alternatives) are themselves discrimination tasks; the probes that degrade (direct deduction, modus ponens, transitive chains) require constructing a conclusion rather than discriminating among options. A

natural hypothesis is that DPO sharpens discrimination at the cost of generation fluency. This could be tested by training a control objective (e.g., SFT on the same pairs) and re-running the expanded probe suite.

**Manifold collapse.** Strong negative margin–capability correlation is consistent with DPO forcing the model onto a "preference manifold" geometrically misaligned with a "truth manifold," collapsing onto a lower-dimensional subspace and losing features required for compositional reasoning. This connects to findings that RLHF can reduce output diversity (Kirk et al., 2024); we observe a related effect at the level of internal probe capability.

**KL-gradient comparability.** The sensitive region near $\beta \approx 0.01$ may mark a regime where the KL-related term becomes comparable in magnitude to the preference term in the update geometry. Below this band, the KL effect may be weak; above it, it may dominate. Future work can test this by tracking per-term gradient norms during training.

## 4.4. Safety Implications

1. **Margin is not a safety proxy.** If margin anticorrelates with capability, standard early-stopping may select worse models.

2. **Benchmarks can mask probe-level degradation.** Surface metrics and aggregate logic margins can improve while specific probe clusters degrade.

3. **Single-seed evaluation is insufficient.** With 2–3 orders of magnitude variance differences across architectures, single runs provide no distributional guarantee.

4. **Training order matters for some capabilities.** Under matched-duration controls, transient high-$\beta$ exposure produces persistent degradation in arithmetic and format in Mistral-7B, but not in logic or sycophancy. Annealing schedules should be validated per-capability.

5. **Capabilities are bundles.** A model can pass an aggregate while failing most constituent probes; probe-level decomposition is essential.

## 4.5. Limitations

- **Scale**: We study 7B models because they permit dense scanning of $\beta$ with tractable compute. Prior work shows capability transitions become sharper with scale (Wei et al., 2022; Caballero et al., 2023), so our 7B results may underestimate alignment-tuning difficulty at frontier scale (Hoffmann et al., 2022).

- **Configuration heterogeneity**: As disclosed in §2.2, our canonical sweeps and our path-dependence experiments use different LoRA configurations ($r = 16, \alpha =$

*Table 8.* Practical checklist for alignment hyperparameter sweeps.

| Recommendations for practitioners |
| --- |
| 1. Do not select $\beta$ using preference margin alone; it can anticorrelate with target capability. |
| 2. Evaluate at multiple abstraction levels (e.g., format *and* probe-cluster decomposition for logic). |
| 3. Run $\geq 3$ seeds near any sensitive $\beta$ region; single runs are unreliable when aggregate variance is high. |
| 4. Test annealing schedules with matched-duration constant-$\beta$ controls before deploying; effects are capability-specific. |
| 5. Evaluate probe-level structure; aggregate metrics can mask systematic sign opposition within a named capability. |

32 vs. $r = 8, \alpha = 16$). The qualitative direction of the main effects reproduces in both, but precise numerical comparison across experiments should be interpreted with this in mind.

- **Corrected path-dependence claim**: An initial A-versus-B comparison suggested logic path dependence but confounded $\beta$-history with training duration. The matched-duration Path C control reported here corrects this: path dependence in Mistral-7B is observed in arithmetic and format, not logic.

- **Probe validation**: We measure log-probability margins on fixed probes. We attempted external validation against ProntoQA (Saparov & He, 2023) as a behavioral logic benchmark; both base models score near chance on this dataset, so it does not validate the log-probability probes for either direction. Validation against logic benchmarks within 7B-base capability remains open (Appendix B).

- **Resolution and coverage**: 13-point grid; 14 expanded logic probes plus 13 in other categories; diagnostic, not comprehensive.

- **Mechanism**: We provide empirical constraints, not internal explanations.

- **Generalization**: We observe co-localized sensitivity at $\beta \approx 0.01$ for these three 7B models under our recipe; whether this threshold generalizes to other scales, optimizers, or DPO variants is open.

## 5. Related Work

**DPO and variants.** DPO (Rafailov et al., 2023) reformulates RLHF as supervised learning. Variants include IPO (Gheshlaghi Azar et al., 2024), KTO (Ethayarajh et al., 2024), ORPO (Hong et al., 2024), and SimPO (Meng et al., 2024). Recent work documents instabilities including length bias (Park et al., 2024; Xu et al., 2024). To our knowledge,

no prior work systematically studies $\beta$ as a control parameter inducing capability-specific responses, nor reports the proxy–capability anticorrelation we document.

**Emergence and metric artifacts.** Wei et al. (2022) document emergent abilities versus scale; Schaeffer et al. (2023) argue these may be metric artifacts. Our expanded-probe analysis identifies a closely related artifact at fixed scale: a 3-probe aggregate can cross zero due to probe-cluster sign opposition without any underlying reasoning transition. We report this as a self-correction and recommend probe-cluster decomposition.

**Sharp transitions in ML.** Sharp transitions have been identified in double descent (Belkin et al., 2019), grokking (Power et al., 2022), scaling laws (Hoffmann et al., 2022), and sudden capability emergence (Chen et al., 2024). Choromanska et al. (2015) connected loss surfaces to spin-glass landscapes. We extend this perspective to post-training alignment, treating $\beta$ as an external control parameter, while emphasizing that the strongest cliff-like behavior we observe is in optimization dynamics (training roughness) and in single-probe sign, not in any aggregate capability metric.

**Goodhart, forgetting, and plasticity.** Proxy-objective decoupling is a known optimization risk (Goodhart, 1984). Gao et al. (2023) characterize reward over-optimization scaling laws; Perez et al. (2022) established that RLHF can induce sycophancy. Askell et al. (2021) and Lin et al. (2024) document capability degradation as an alignment cost. Dohare et al. (2024) demonstrate that networks can lose the ability to learn after certain training regimes; we observe a related capability-specific persistence under DPO. Our results extend this literature with a controlled matched-duration test, a quantitative proxy–capability anticorrelation, and a probe-cluster decomposition.

## 6. Conclusion

Sweeping $\beta$ as a control parameter under our DPO recipes reveals that the standard tuning pipeline (maximize the preference margin, validate on aggregate benchmarks) can select capability-impaired models. The DPO preference margin can strongly anticorrelate with logic-probe capability ($r = -0.91$ for LLaMA-2-7B logic); under matched-duration controls, transient high-$\beta$ exposure induces persistent degradation in arithmetic and format, but not in logic or sycophancy; and an expanded 14-probe logic suite reveals that aggregate margin is structured by probe clusters that move in opposite directions under DPO, with 13 of 14 probe signs stable across seeds and $\beta$ values.

The strongest supported claims concern proxy decoupling, capability-specific path dependence, and probe-cluster sign

structure. We propose a probe-resolved evaluation methodology: (1) map capability-vs-$\beta$ structure with multiple probes per capability; (2) validate with multiple seeds in sensitive regions; (3) use matched-duration controls before deploying annealing schedules; (4) monitor probe-cluster signs rather than only aggregate margins.

The practical question is not "what $\beta$ maximizes margin?" but "can we certify the model remains capability-ordered under realistic training paths, at the level of probe clusters rather than aggregate scores?"

## Impact Statement

This work advances scientific understanding of alignment dynamics by empirically characterizing non-monotonic and capability-specific effects of DPO. Its primary contribution is methodological: providing diagnostics for more reliable evaluation of alignment hyperparameters.

The findings may have positive societal impact by informing safer deployment practices, highlighting failure modes where standard alignment metrics are misleading, and encouraging more thorough evaluation. The work does not introduce new model architectures, datasets, or deployment mechanisms and does not directly enable harmful applications beyond those already associated with large language models.

As with all research on model alignment, there is a dual-use consideration: insights into optimization-induced failures could be misused to target brittle regions of hyperparameter space. We believe transparency about these dynamics is essential for improving robustness and safety, and that the benefits of identifying and mitigating these risks outweigh potential misuse.

## Reproducibility Statement

**Models.** All experiments use publicly available 7B-scale checkpoints: Mistral-7B (`mistralai/Mistral-7B-v 0.1`), LLaMA-2-7B (`NousResearch/Llama-2-7 b-hf`), and Qwen-1.5-7B (`Qwen/Qwen1.5-7B`).

**Protocols.** The canonical $\beta$ sweep trains each $\beta$ point from a fresh base checkpoint. The matched-duration path-dependence test (Paths A, B, C) deliberately varies $\beta$-history (§3.4). Full training hyperparameters, including the two LoRA configurations used across experiment families, are in §2.2.

**Evaluation.** Capability is measured by length-normalized log-probability margins on fixed probe sets. Positive margin indicates that the correct continuation is preferred. For GSM8K, $\beta = 0.002$ is treated as missing data due to a degenerate zero score under our evaluation pipeline.

**Artifacts.** We release a public repository containing all hyperparameter settings, runnable scripts, raw per-run JSON outputs, expanded-probe definitions, and plotting code sufficient to reproduce every figure and table: https://github.com/mpollanen/dpo-capability-traps.

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

## A. Expanded Probe Suite

### A.1. Full Probe Text

We list all probes used in the expanded suite (27 probes total). For each probe we give name, difficulty label, the reasoning skill tested, the prompt, the correct continuation, and the distractor. Margins are length-normalized log-probability differences between the correct and distractor continuations.

**Logic (14 probes).** *L1 (simple direct inference).*

- syllogism_1. "All bloops are razzles. All razzles are flerms. Are all bloops flerms? (Yes/No)" Correct: Yes. Distractor: No. Tests: $A \rightarrow B$ and $B \rightarrow C$ implies $A \rightarrow C$.

- `transitive_simple`. "All wugs are daxes. All daxes are blickets. Is a wug a blicket? (Yes/No)" Correct: Yes. Distractor: No. Tests: transitive category membership.

- `modus_ponens`. "If it is raining, the ground is wet. It is raining. Is the ground wet? (Yes/No)" Correct: Yes. Distractor: No. Tests: direct modus ponens.

- `simple_negation`. "No zorps are flimbles. Kiki is a zorp. Is Kiki a flimble? (Yes/No)" Correct: No. Distractor: Yes. Tests: negation in conclusion.

*L2 (moderate / requires care).*

- `syllogism_2`. "All glims are zorks. Some zorks are blats. Are some glims blats? (Yes/No)" Correct: No. Distractor: Yes. Tests: invalid syllogism (undistributed middle).

- `ordering`. "Alice is taller than Bob. Bob is taller than Carol. Who is shortest?" Correct: Carol. Distractor: Bob. Tests: 3-entity transitive ordering.

- `converse_error`. "If an animal is a dog, it has four legs. This animal has four legs. Is it necessarily a dog? (Yes/No)" Correct: No. Distractor: Yes. Tests: affirming the consequent (invalid).

- `inverse_error`. "If it snows, the schools close. It did not snow. Can we conclude the schools are open? (Yes/No)" Correct: No. Distractor: Yes. Tests: denying the antecedent (invalid).

- `modus_tollens`. "If the battery is charged, the light turns on. The light did not turn on. Is the battery charged? (Yes/No)" Correct: No. Distractor: Yes. Tests: modus tollens.

- `some_are_not`. "Some plonks are greebles. Some plonks are not greebles. Are all plonks greebles? (Yes/No)" Correct: No. Distractor: Yes. Tests: existential negation.

*L3 (multi-step / harder).*

- `chain_4step`. "All blerps are snorgs. All snorgs are twills. All twills are praxes. All praxes are vonks. Is a blerp a vonk? (Yes/No)" Correct: Yes. Distractor: No. Tests: 4-step transitive chain.

- `chain_with_negation`. "All frobs are gleeks. All gleeks are murns. No murns are pliffs. Is a frob a pliff? (Yes/No)" Correct: No. Distractor: Yes. Tests: transitive chain with negation at one step.

- `ordering_4entity`. "Dave is shorter than Carol. Alice is taller than Bob. Bob is taller than Carol. Who is the shortest?" Correct: Dave. Distractor: Carol. Tests: 4-entity ordering with reordering.

- `contrapositive_chain`. "If a shape is a square, then it is a rectangle. If a shape is a rectangle, then it has four sides. This shape does not have four sides. Is it a square? (Yes/No)" Correct: No. Distractor: Yes. Tests: contrapositive reasoning through a chain.

**Other categories.** Arithmetic (3 probes; `carry_add`, `carry_add_2`, `borrow_sub`), Format (4 probes; `json_simple`, `json_key`, `strict_bool`, `strict_json`), Sycophancy (3 probes; `flat_earth`, `bad_math`, `cheese_moon`), Negation (3 probes; `not_fruit`, `no_blue`, `not_bird`). Full text of all probes is in the reproducibility archive.

### A.2. Multi-Seed Sign Stability

Table 9 summarizes per-probe sign across 3 seeds at 3 $\beta$ values in Mistral-7B.

*Table 9.* Per-probe sign stability across 3 seeds $\times$ 3 $\beta$ values ($\beta \in \{0.006, 0.01, 0.02\}$, 9 conditions per probe) in Mistral-7B, expanded logic suite. "9-cond. signs" counts how many of the 9 conditions produce the indicated sign.

| Cluster | Probes | 9-cond. signs |
|---|---|---|
| Always negative (direct inference) | `syllogism_1`, `ordering`, `transitive_simple`, `modus_ponens`, `chain_4step` | 5 probes; 45/45 negative |
| Always positive (fallacy detection / discrimination) | `syllogism_2`, `simple_negation`, `converse_error`, `inverse_error`, `some_are_not`, `chain_with_negation`, `ordering_4entity`, `contrapositive_chain` | 8 probes; 72/72 positive |
| Mixed sign | `modus_tollens` (flips at $\beta = 0.02$, seed 1) | 1 probe; 1 flip in 9 |

## B. ProntoQA Validation Attempt

To address whether log-probability probes track behavioral logical reasoning, we evaluated all Mistral-7B and LLaMA-2-7B canonical-sweep checkpoints on ProntoQA (Saparov & He, 2023) (200 validation examples). Results (200 examples per condition):

Both base models perform at or near chance on ProntoQA under log-probability scoring (Mistral-7B 0.515, LLaMA-2-7B 0.500). The task exceeds the capability of these 7B base checkpoints. Consequently, ProntoQA in this setting neither

*Table 10.* ProntoQA results across the canonical sweep. LP_acc: log-probability accuracy. Gen_acc: parsed generation accuracy. "Sweep" rows give the spread observed across $\beta \in [0.001, 0.05]$.

| Model | $\beta$ | LP_acc | Gen_acc | Parse rate |
|---|---|---|---|---|
| Mistral-7B | base | 0.515 | 0.485 | 1.000 |
| Mistral-7B | sweep | $\sim$0.50 | 0.00–0.17 | 0.00–0.33 |
| LLaMA-2-7B | base | 0.500 | 0.015 | 0.020 |
| LLaMA-2-7B | sweep | $\sim$0.515 | $\sim$0.000 | $\sim$0.005 |

validates nor invalidates the log-probability probes used in the main paper; it does not provide ground truth at the operating point where the probes show variation. We report this honestly as a null validation result and identify validation against logic benchmarks within 7B-base capability as future work.

## C. Structural Collapse

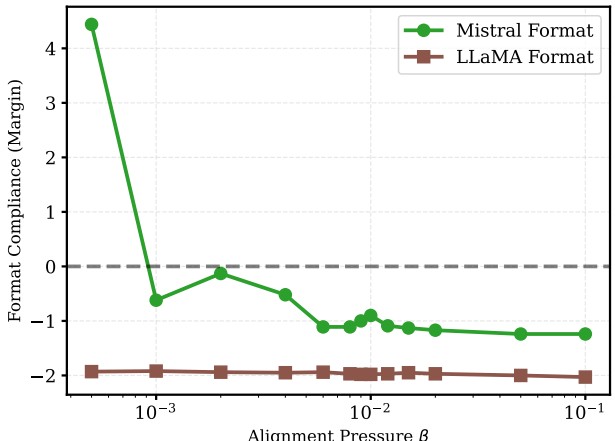

*Figure 6.* **Structural collapse.** Mistral-7B format integrity ($+4.44$ at the lowest $\beta$) degrades with $\beta$. LLaMA-2-7B begins with a negative format margin and never resolves.

Format compliance probes low-level structural priors (Figure 6). Mistral-7B begins with high integrity ($+4.44$ at $\beta = 0.0005$) which erodes as $\beta$ increases, aligning with our matched-duration path-dependence result for format (§3.4). LLaMA-2-7B begins with broken priors (format margin $\approx -2$ under our sweep) that never resolve, consistent with its selective response mode.

## D. Cross-Architecture Sensitivity Scan

A natural question is whether the sensitive region observed in Mistral-7B reflects a peculiarity of that architecture or a broader phenomenon. Across three 7B families under the same sweep, only Mistral-7B exhibits 3-probe-aggregate sign changes; the other architectures show qualitatively different behavior near $\beta \approx 0.01$. Mistral-7B produces ag-

gregate sign changes with elevated seed variance; LLaMA-2-7B responds selectively (sycophancy and format change while logic remains essentially frozen); Qwen-1.5-7B absorbs pressure smoothly (continuous trade-offs without sharp discontinuities in our probe set).

We additionally ran an auxiliary associative-logic diagnostic on Mistral-7B and Qwen-1.5-7B (seed 1) under a faster training schedule (100 steps, lr $= 2 \times 10^{-4}$) and release the raw outputs in the reproducibility archive. Under this probe family, Mistral-7B exhibits a sharp increase in associative margin between $\beta = 0.009$ and $\beta = 0.010$ ($+175\%$), co-localizing with the canonical roughness discontinuity in Table 1, while Qwen-1.5-7B varies smoothly across the grid.

## E. 1000-Step Extended Sweep

Per-point results for the Mistral-7B sweep at 1000 steps (LoRA $r = 8$, $\alpha = 16$; seed 1). At $\beta = 0.010$, the two syllogism probes reverse sign relative to neighbors (syllogism_1 becomes positive at $+2.53$ while syllogism_2 becomes negative at $-5.19$) and the preference margin spikes to 41.1 at the same point where aggregate logic is most negative ($-0.96$).

*Table 11.* Mistral-7B extended sweep to 1000 steps. Bold: aggregate logic positive.

| $\beta$ | Logic | Arith | Syll_1 | Syll_2 | Margin |
|---|---|---|---|---|---|
| 0.004 | $-0.64$ | $+3.84$ | $-5.71$ | $+5.27$ | 11.9 |
| **0.006** | **$+0.23$** | $+4.53$ | $-4.95$ | $+4.56$ | 15.8 |
| **0.008** | **$+0.34$** | $+4.38$ | $-4.54$ | $+5.69$ | 18.9 |
| 0.010 | $-0.96$ | $+4.33$ | $+2.53$ | $-5.19$ | **41.1** |
| 0.012 | $-0.29$ | $+4.31$ | $-0.80$ | $+1.77$ | 24.1 |
| 0.015 | $-0.52$ | $+4.87$ | $-0.67$ | $+1.50$ | 22.5 |
| 0.020 | $-0.73$ | $+4.44$ | $-2.73$ | $+2.88$ | 26.2 |

## F. Learning-Rate Stress Test (LLaMA)

LLaMA-2-7B logic-probe margins for the learning-rate stress test reported in §3.2. We evaluated three learning rates ($1 \times 10^{-5}$, $5 \times 10^{-5}$ canonical, $2 \times 10^{-4}$) at $\beta \in \{0.006, 0.01, 0.02\}$, seed 1. All 9 conditions yielded negative aggregate logic margins clustered tightly around $-0.97$, with total spread (maximum minus minimum across the grid) of 0.125. Increased step size therefore does not change the sign of the logic profile or recover positive logic at any tested $\beta$. Per-condition raw values are released in the reproducibility archive.

## G. Hardware and Software Configuration

Most experiments were run on a RunPod L4 (24 GB): the canonical $\beta$ sweep, multi-seed variance, architecture

comparison, learning-rate stress test, and expanded probe evaluations. The matched-duration path-dependence experiments (Paths A, B, and C, run together on identical hardware to support the paired $t$-test) and the 1000-step extended sweep were run on a RunPod L40S. Software stack: `trl` 0.12.2, `transformers` 4.46.3, `peft`, `bitsandbytes`, `accelerate`, CUDA 12.8, and PyTorch 2.4.1. Models were loaded in 4-bit quantization for generation; full-precision training under the LoRA configurations stated in §2.2.

