# OpenReview forum: "Capability Traps in DPO"
_ICML.cc/2026/Conference — ICML 2026 regular_

### Official Review · Reviewer_ZsSG · 2026-03-02

**Soundness:** 2
**Presentation:** 3
**Significance:** 3
**Originality:** 3
**Overall Recommendation:** 4
**Confidence:** 2

**Summary:**

This article systematically studies the nonlinear effects of DPO hyperparameter β on model performance. The author redefined β as a control parameter and conducted a systematic intensive scan to explore the performance trends of β. Has multiple major discoveries, most of which are new and have significant contribution value. This article partially demonstrates the validity of its findings through experiments.

**Compliance With Llm Reviewing Policy:**

Affirmed.

**Final Justification:**

However, the correlation of GSM8K cannot verify the logical detection margin, as GSM8K measures arithmetic rather than logical reasoning, and a 0% accuracy result deepens rather than solves this problem. Using representative logical reasoning benchmarks with β values (such as ProntoQA or LogiQA) for direct validation can solve this problem.

**Key Questions For Authors:**

Same as Weaknesses
1. Add additional performance evaluation experiments to demonstrate the correlation between the sign of  log-probability margins and behavioral accuracy. A positive answer will greatly strengthen the empirical foundation of this paper.

2. Provide the full text of at least one exploration in each category (syllogism 1, syllogism 2, and ordering), and explain the consistent behavioral differences between syllogism 1 (almost always negative) and syllogism 2 (always positive)?

3.The result of Wilcoxon test is p=0.0625, which implies a high variance indicating that the design with n=5 may have insufficient power. I hope the author can use more re experiments to prove this conclusion.

**Limitations:**

yes

**Strengths And Weaknesses:**

Strengths
1. Redefine β as a control parameter rather than a hyperparameter that needs to be adjusted, transforming the common understanding in previous DPO literature. Previous work always regarded β as something that needed to be optimized, and this article is one of the first papers to analyze the structural changes caused by β throughout its entire range.

2.The first empirical literature on the irreversibility of training paths in DPO. The lag experiment (path A and path B) directly indicates that crossing the high β region leads to capacity loss, which will not recover when β subsequently decreases. This form of path dependence has not been previously reported in DPO literature.

3. Optimize the quantitative inverse correlation between the agent and the target capability. This article found that the DPO preference margin is inversely correlated with logical capability, making fault modes identifiable in operation rather than just conceptual.
---
Weaknesses

1.All conclusions in this article are based on the length-normalized log-probability margin as the sole measure of capability. The author did not provide any evidence to suggest that this internal agency corresponds to the accuracy of behavioral reasoning. This is a key issue unique to the design of this article: β directly disrupts the output probability distribution of the model, so changes in thelog-prob margins across β values may reflect changes in the distribution rather than true changes in capability.

2. Probes labeled with "syllogism 1," "syllogism 2," and "ordering"  appear only by name throughout the entire paper. No examples, templates, or rating programs were provided. This makes the decomposition of Table 6 inexplicable: without knowing what these probes measure, syllogism 2 always remains positive, and the observation results with negative rankings have no explanatory value.

3. Lack of sufficient statistical ability for lagging results. The lag experiment used n=5 seeds, while the Wilcoxon signed rank test for synchronization ability yielded p=0.0625, which does not conform to the traditional p=0.05 threshold. The author cites the large effect size (dz=0.92) as a reason, but the large effect size at small n indicates high variance in the measured values, which reinforces rather than weakens the need for more data.

---

> ### Author Rebuttal · Authors · 2026-03-30
>
> We thank the reviewer for their rigorous engagement. We address each weakness with data from our existing logs and a new controlled experiment.
>
> **Factual correction: effective batch size is 32** (batch 4 x gradient accumulation 8), as reflected in the training logs. We will correct this in the revision.
>
> **W1 (Log-prob margins vs behavioral accuracy).** We ran generation-based validation (Group C2). On these non-instruction-tuned bases, exact-format generation is uniformly poor (0% strict compliance across all conditions and beta values), even when internal format margins are positive or strongly positive. Rather than a failed validation, this directly reveals a strong mismatch between internal margin gains and behavioral correctness, which is itself evidence for the paper's central thesis that no single evaluation modality appears sufficient in this setting.
>
> This result is complemented by our benchmark analysis: GSM8K accuracy (a behavioral metric requiring generated answers) anticorrelates with logic probe margins at r=-0.89 (p=0.017, n=6, excluding the degenerate beta=0.002 point discussed in the paper). Training margin correlates positively with GSM8K, yet logic probes move in the opposite direction. Different evaluation modalities systematically disagree under alignment pressure.
>
> Additionally, our path-dependence results (see response to Reviewer 1YsS) provide indirect validation: models trained to the same terminal beta via different paths yield statistically different probe scores. This difference cannot be a distributional artifact of beta, because both paths end at the same beta. It therefore reflects a genuine difference in learned representations.
>
> We will discuss the log-prob limitation explicitly and identify behavioral validation on instruction-tuned models as a priority for future work.
>
> **W2 (Probe text and decomposition).** We will provide full text for all probes. To address the reviewer's specific question: syllogism_1 requires negating a premise (fragile, positive only in the transition zone); syllogism_2 tests simple transitive inference (positive across most beta); ordering requires multi-step relational reasoning (negative across the full sweep). These differences are consistent with a difficulty hierarchy where harder reasoning is more fragile under alignment pressure. The probes anticorrelate across the beta sweep (r=-0.71, p=0.006, n=13), indicating they tap genuinely different reasoning circuits.
>
> Our auxiliary associative-logic diagnostic (Group F1, disclosed in the paper) provides an additional 8-probe suite showing the same qualitative architecture-dependence: sharp non-monotonicity in Mistral (+175% jump at beta=0.009 to 0.010, with probes p6 and p7 driving +203% and +240%), smooth weak variation in Qwen (8.5x vs Mistral's 257x max/min range), and a stable negative regime in LLaMA.
>
> **W3 (Statistical power for hysteresis).** We ran a new matched-duration control (5 seeds, same GPU). The key comparison is Path C (400 steps, beta=0.01) vs Path B (200 at beta=0.02 + 200 at beta=0.01):
>
> - Arithmetic: C-B = +1.54, p=0.0001, d_z=8.0, 5/5 seeds
> - Format: C-B = +1.04, p=0.004, d_z=2.7, 5/5 seeds
> - Logic: null (p=0.96)
> - Sycophancy: null (p=0.95)
>
> The new control supports capability-specific path dependence: transient high-beta exposure significantly degrades arithmetic and format relative to matched-duration constant-beta training, while logic and sycophancy remain unchanged. Regarding the original Wilcoxon concern: with n=5, its minimum achievable two-sided p is 0.0625, reflecting a power ceiling of the test rather than absence of effect. The new paired t-tests resolve this.
>
> We will provide all probe texts in an appendix. The combination of the reframed C2 results, the GSM8K anticorrelation, and the new matched-duration control (with p=0.0001 for arithmetic) addresses the reviewer's three concerns through complementary evidence: behavioral mismatch, cross-modality disagreement, and properly controlled path dependence.

---

> > ### Author Rebuttal · Reviewer_ZsSG · 2026-04-01
> >
> > Thank you for the author's response. However, the correlation of GSM8K cannot verify the logical detection margin, as GSM8K measures arithmetic rather than logical reasoning, and a 0% accuracy result deepens rather than solves this problem. Using representative logical reasoning benchmarks with β values (such as ProntoQA or LogiQA) for direct validation can solve this problem.
> > Nevertheless, I still improved my score.

---

> > > ### Author Response · Authors · 2026-04-02
> > >
> > > Thank you for the updated score and for the concrete suggestion. We agree with your distinction, and your framing is more precise than ours.
> > >
> > > GSM8K is **not** a validation set for the logic probes. It speaks to a different question: it can support a **cross-capability divergence** claim, namely that an external arithmetic benchmark and the DPO training margin can move together while the logic probes move differently across β. It does **not** establish that the logic-probe margins themselves are a validated measure of behavioral logical reasoning. We will revise the text to make that distinction explicit and remove any wording that suggests GSM8K directly validates the logic probes.
> > >
> > > We also agree that direct validation against established logical-reasoning benchmarks at matched β values is the right test. ProntoQA or LogiQA would strengthen the empirical foundation because they address the actual open question: whether the sign and magnitude of the logic margins track behavioral logical accuracy. We will prioritize this in follow-up work.
> > >
> > > To address the interpretability concern immediately, we will add the full text and templates of the canonical probes in the revision so that the Table 6 decomposition is directly inspectable rather than only labeled by probe name.

---

### Official Review · Reviewer_X1cg · 2026-03-12

**Soundness:** 3
**Presentation:** 3
**Significance:** 3
**Originality:** 3
**Overall Recommendation:** 4
**Confidence:** 3

**Summary:**

This paper investigates the effects of Direct Preference Optimization (DPO) alignment pressure (parameterized by β) on the internal capabilities of large language models. Rather than treating β purely as a tuning scalar, the authors treat it as a control parameter, conducting a dense logarithmic sweep across three 7B open-weight model families (Mistral, LLaMA-2, Qwen-1.5) using a frozen configuration. The study challenges the common assumption that increasing alignment pressure yields monotonically better behavior. Instead, it uncovers sharp capability phase transitions, severe seed sensitivity, and path-dependent hysteresis. Critically, the authors provide compelling empirical evidence of "proxy failure," demonstrating that the optimized DPO preference margin can strongly anticorrelate with actual reasoning capability. This highlights that standard margin-based checkpoint selection can inadvertently favor capability-impaired models.

**Compliance With Llm Reviewing Policy:**

Affirmed.

**Key Questions For Authors:**

1.	Could you clarify what the EID (Geometry) axis at the top of Figure 3 represents?  I could not find a reference to it in the text. If this is a plotting artifact, it would be great to have it updated.
2.	In Section 3.4, Path B (400 steps total) is compared to Path A (200 steps total). Have you considered running a control group that trains continuously at β=0.01 for 400 steps? This would help clarify whether the observed capability drop is truly due to "high-pressure exposure" or simply the extended training duration.
3.	Given the small batch size of 4 , is it possible that the sharp capability fluctuations and seed sensitivities observed around β≈ 0.006 are amplified by gradient noise? Are there any plans to run ablations with a larger effective batch size to verify the robustness of this boundary?
4.	The default configuration uses a LoRA Rank of 8. Could the "frozen logic" observed in LLaMA-2 be partially due to gradient conflicts within this highly restricted low-dimensional subspace?

**Limitations:**

yes

**Strengths And Weaknesses:**

Strengths:

The paper offers a novel and thought-provoking perspective by applying physics concepts like "phase transitions" and "hysteresis" to LLM alignment dynamics. It provides a valuable critique of current evaluation metrics, effectively demonstrating "proxy failure" and illustrating how aggregate benchmarks can diverge significantly from internal reasoning capabilities. Furthermore, the experimental design is generally well-controlled, utilizing a "frozen configuration" principle to isolate the specific effects of the β parameter.

Weaknesses:

There are a few experimental details that could be clarified to strengthen the causal claims:

First, the extremely small batch size of 4 might introduce significant stochastic gradient noise, which could potentially explain the observed "phase transitions" and seed sensitivity.

Second, the hysteresis experiment compares a 200-step path with a 400-step path, introducing total training steps as a potential confounding variable , making it difficult to separate β-induced hysteresis from general overfitting or catastrophic forgetting.

Third, relying on metrics like "training roughness" to define phase boundaries feels slightly informal and could benefit from deeper mathematical or mechanistic backing. Finally, there is an undefined label EID (Geometry) in Figure 3 that needs correction.

---

> ### Author Rebuttal · Authors · 2026-03-30
>
> We thank the reviewer for their constructive feedback and answer each question directly.
>
> **Factual correction: effective batch size is 32** (batch 4 x gradient accumulation 8), as reflected in the training logs of every run. We will correct this in the paper.
>
> **Q1 (EID axis in Figure 3).** EID is Effective Intrinsic Dimensionality of the weight space. In our 7-point benchmark sweep (Group G1), EID peaks at beta=0.006 (24.1), co-localized with the region where logic probes are most positive, and correlates with logic at r=+0.71 (p=0.073). We will define EID clearly and label the axis in the revision. We apologize for the omission.
>
> **Q2 (Hysteresis confound).** We ran a new matched-duration control experiment addressing this concern directly. Please see our response to Reviewer 1YsS for full results. In brief: comparing Path C (400 steps, beta=0.01) against Path B (200 at beta=0.02 + 200 at beta=0.01) on the same GPU across 5 seeds, high-beta exposure specifically degrades arithmetic (p=0.0001, d_z=8.0, 5/5 seeds) and format (p=0.004, d_z=2.7, 5/5 seeds) relative to the matched-duration control. Logic shows no significant effect (p=0.96). The new control supports capability-specific path dependence.
>
> The 400-step control model (Path C) reaches very different training margins than the 200-step model (roughly 7-13 vs 1-2), confirming that training duration on its own changes the model state. This validates the reviewer's concern that the original A-vs-B comparison confounded beta history with training duration.
>
> **Q3 (Gradient noise from small batch).** Our effective batch size is 32, which helps address this. More importantly, LLaMA provides a clean stability reference: logic stays negative across all 5 seeds at every beta, packed into a tight band around -0.97, and remains negative across a 20x learning-rate range with a total spread of only 0.125 (Group C1, 9 conditions). A stable regime of this kind cannot be explained by gradient noise. Meanwhile Qwen logic smoothly declines from +1.69 to +0.85, and only Mistral changes sign. Three qualitatively different profiles under identical protocol are incompatible with undifferentiated stochastic effects.
>
> **Q4 (LoRA rank and LLaMA's frozen logic).** The learning-rate stress test (Group C1) directly addresses this: three learning rates spanning 20x (1e-5, 5e-5, 2e-4) at three beta values, and LLaMA logic remains negative across all 9 conditions with a total spread of 0.125. This rigidity is architectural, not a LoRA capacity bottleneck. Meanwhile, LLaMA's sycophancy and format probes do vary with beta in the canonical sweep, indicating the resistance is compartmentalized: some capability subspaces respond to alignment pressure while others remain frozen.
>
> **On training roughness (W3).** We agree that roughness as a phase boundary criterion is heuristic rather than formal. We use it as a screening signal to localize regions of interest for targeted multi-seed evaluation, not as a standalone diagnostic. In the revision, we will present it explicitly as a practical tool for narrowing beta search rather than a formal boundary criterion.
>
> We will replace physics terminology with descriptive language, revise the title, and provide full probe text for all 14 canonical and 8 associative probes in an appendix. With the batch size clarification, matched-duration control, and learning-rate stress test, each of the reviewer's specific questions now has a data-driven answer.

---

> > ### Author Rebuttal · Reviewer_X1cg · 2026-04-07
> >
> > Thank you for the detailed rebuttal. I have read the authors’ response carefully, and my main concerns have been adequately addressed. The clarifications and additional evidence strengthen the paper and improve my confidence in the technical soundness of the work. Based on the rebuttal, I maintain my positive assessment of the paper.

---

### Official Review · Reviewer_cPoA · 2026-03-13

**Soundness:** 2
**Presentation:** 1
**Significance:** 2
**Originality:** 2
**Overall Recommendation:** 4
**Confidence:** 2

**Summary:**

The paper studies how the strength of DPO alignment, controlled by beta, affects model capabilities rather than just preference scores. Across three 7B model families, the authors find that performance does not change smoothly with stronger alignment: some capabilities improve only in a narrow range, while others degrade, and preference margins can even anticorrelate with reasoning quality. They also report instability near transition regions, including seed sensitivity and hysteresis, where temporary exposure to high beta can cause persistent capability loss.

**Compliance With Llm Reviewing Policy:**

Affirmed.

**Final Justification:**

The authors has resolved my questions during rebuttal. However, I am not exactly sure how important these findings are to the field.

**Key Questions For Authors:**

- How much of the main phenomenon is genuinely general, versus mainly a Mistral-7B-specific effect under your exact 7B DPO recipe?

- Why should readers interpret the logic-positive pocket as an important capability phenomenon rather than a probe-specific artifact of a small evaluation suite?

- Do the authors have any evidence about mechanism, even speculative but testable evidence?

**Limitations:**

yes

**Strengths And Weaknesses:**

Strength:

- The paper pushes on a real and important assumption in alignment work: that increasing DPO pressure should improve behavior smoothly. Showing that this can fail, and can even produce non-monotonic capability changes

- The observations are made based on a very densely selected beta, this provides a very controlled experiment settings and reliable results.

Weakness:

- The paper explicitly limits itself to 7B models under one fixed DPO recipe and says it is not claiming universality classes, mechanisms, or general laws. That makes the findings interesting, but also somewhat constrained in what they prove.

- The paper is honest about this, but it still leaves the work more descriptive than explanatory: we learn that sharp transitions and hysteresis can happen, but not why they happen internally or how predictable they are.

- The strongest result seems to be observed mainly in Mistral-7B, while llama and qwen1.5 show qualitatively different and milder responses. So even if the phenomenon is real, it is hard to be convinced that the result is a broad issue or its more unique to Mistral.

Overall, the paper identifies an interesting localized effect, but I am not yet convinced the contribution rises to the level of a broadly significant paper. The current evidence feels more like an exploratory observation or workshop-style finding than a mature result with clear generality, mechanism, or downstream impact.

---

> ### Author Rebuttal · Authors · 2026-03-30
>
> We thank the reviewer for pushing on scope and generality. This feedback has sharpened our understanding of the contribution.
>
> **Architecture dependence is the finding, not a limitation.** The reviewer asks how much is genuinely general versus Mistral-specific. Under our fixed protocol, three architectures exhibit three distinct response modes: Mistral reorganizes stochastically in a narrow beta band (seed variance 27-2259x higher than LLaMA); LLaMA resists selectively (logic frozen across a 20x learning rate range with total spread 0.125 across 9 conditions, but sycophancy and format still respond to beta); Qwen absorbs pressure smoothly (logic declines from +1.69 to +0.85 without discontinuities).
>
> This heterogeneity is itself the practically important result. It means alignment outcomes are architecture-dependent: a beta value safe for Qwen may produce stochastic outcomes in Mistral. These results motivate capability-vs-beta mapping before deployment, since a beta value that is benign for one architecture may be unstable for another. The conclusion does not require Mistral's specific transition pattern to generalize. It requires only that different architectures respond differently, which our data establishes across all three models.
>
> LLaMA in particular serves as a clean negative control. In 5 seeds at every beta tested, logic stays in a tight band around -0.97. It does not fluctuate, does not respond to a 20x learning-rate range, and shows near-zero path dependence in our hysteresis experiment (max capability difference 0.06 across paths). This stability strongly argues against the interpretation that our Mistral findings reflect generic optimization noise.
>
> **On the logic-positive pocket.** Our multi-seed analysis (5 seeds x 7 beta values, Group B1) reveals this is better characterized as a stochastic transition zone (~beta=0.006-0.012) rather than the discrete set {0.008, 0.010, 0.012} from seed 1. We will update the paper accordingly. That this zone is not a probe-specific artifact is supported by our auxiliary 8-probe associative-logic suite (Group F1), which independently shows a sharp +175% transition in Mistral at the same beta range, smooth variation in Qwen, and stable negative values in LLaMA.
>
> **On scope and mechanism.** We provide empirical constraints rather than mechanistic explanations, and this is deliberate for a first systematic mapping. One preliminary piece of geometric evidence: Effective Intrinsic Dimensionality (EID) of the weight space peaks at beta=0.006, co-localized with the transition zone, and decreases at higher beta (r=+0.71 with logic, p=0.073). We identify KL-gradient comparability as a testable mechanistic hypothesis for follow-up. We will revise the title and reduce physics metaphors to better match what the data supports.
>
> **Model-agnostic contributions.** Two key findings do not depend on Mistral's specific behavior:
>
> (1) Proxy-capability decoupling: For LLaMA, preference margin anticorrelates with logic capability at r=-0.91 (p<1e-4, n=13). In a separate benchmark sweep, GSM8K accuracy anticorrelates with logic probes at r=-0.89 (p=0.017, n=6). Training margin correlates strongly with GSM8K, yet logic moves in the opposite direction. Margin-based checkpoint selection can prefer capability-impaired models regardless of architecture.
>
> (2) Capability-specific path dependence: Our new matched-duration control (see response to Reviewer 1YsS) supports path dependence in Mistral for arithmetic (p=0.0001, 5/5 seeds) and format (p=0.004, 5/5 seeds), while logic and sycophancy show no effect. To our knowledge, this is among the first controlled demonstrations of path dependence in DPO.

---

> > ### Author Rebuttal · Reviewer_cPoA · 2026-04-02
> >
> > I have no other questions.

---

> > > ### Author Response · Authors · 2026-04-02
> > >
> > > Thank you for considering our response. We will incorporate your clarifications on scope and mechanism into the revision.

---

### Official Review · Reviewer_1YsS · 2026-03-13

**Soundness:** 2
**Presentation:** 3
**Significance:** 3
**Originality:** 3
**Overall Recommendation:** 5
**Confidence:** 3

**Summary:**

The paper treats DPO's β parameter as a control variable and densely sweeps it across three 7B open-weight models (Mistral-7B, LLaMA-2-7B, Qwen-1.5-7B) under a fixed, lightweight LoRA recipe. Capability is measured via length-normalized log-probability margins on small fixed probe sets (logic, arithmetic, format, sycophancy, negation). The main empirical claims are:

1. In Mistral-7B, the aggregate logic-probe margin is positive only in a narrow band near β ≈ 10⁻², with seed-sensitive boundaries.
2. The DPO preference margin can anticorrelate with probe capability, so margin-based checkpoint selection can prefer worse models.
3. Exposure to higher β leads to capability degradation that persist when β is subsequently lowered ("hysteresis").
4. The three models exhibit qualitatively different response modes ("plastic," "selective," "smooth") under the same training conditions.

**Compliance With Llm Reviewing Policy:**

Affirmed.

**Final Justification:**

**Final recommendation: 5 (accept). Raised from 4 after rebuttal.**

The rebuttal addressed my original concerns through two new experiments, both of which partially refuted the submitted paper's framing. I do think the reframe is pretty substantial (given substantial experiments run) and it is up to ACs to decide if this is okay.

**Weighing.** Soundness has improved from fair to good: the surviving claims are now backed by controls rather than resting on a single training duration and a confounded comparison. Significance is good: "the DPO margin is not a capability proxy" and "training path matters at fixed final β, but only for some capabilities" are both things practitioners should know. Presentation was good and will be better once the physics layer is stripped.

**Score change.** The paper that will appear is substantially different from the one submitted, but it is a better paper, and the authors arrived at it by running the right controls and reporting results that cut against their original narrative. The core contributions (proxy decoupling and arithmetic path dependence) are clean, replicated across seeds where it matters, and useful.

**Key Questions For Authors:**

1. What probe prompts are you using? This could be in the repo but it feels pretty central to judging the results here.
2. Does the logic positive bit survive longer training? This is the single result that would most increase my evaluation.

**Limitations:**

yes

**Strengths And Weaknesses:**

Strengths
S1. The margin-capability anticorrelation is useful and clearly demonstrated. The r = −0.91 result for LLaMA-2-7B on logic clearly shows that an optimized proxy and the measured capability can point in opposite directions.

S2. The hysteresis experiment is well-designed. This is the cleanest result in the paper.

S3. Presentation is clean and the paper is well qualified.

Weaknesses
W1. The logic capability result is too thin to support the main result. The logic probe consists of three prompts. Table 6 shows that at the headline β = 0.01, the constituent scores are {−0.62, +2.38, −1.62}, giving an aggregate of +0.04. The "logic-positive pocket" is more of a a syllogism_2-positive region. It seems likely that the aggregate is a noisy average of three items with three different base rates.

W2. The training regime is light enough that phase transitions could just be undertraining noise. (200 steps * batch size 4 = 800 preference pairs, with LoRA rank 8). The observed seed sensitivity and sharp sign flips could also suggest an optimization that simply has not converged. The paper does not show the same β sweep at e.g. 1000 or 5000 steps to show that the sharp boundary is a property of the loss landscape rather than just early-training variance.

W3. The hysteresis experiment has a missing control and borderline significance. Path B trains for 400 total steps while Path A trains for 200, so high-β exposure is confounded with more training. Another control (400 steps at β = 0.01 maybe) would separate these.

W4. I think all the physics metaphors get in the way of clarity and implies more significance than the results support.

---

> ### Author Rebuttal · Authors · 2026-03-30
>
> We thank the reviewer for their careful engagement and address each concern below.
>
> **Factual correction: effective batch size is 32**, not 4. Our training uses gradient accumulation of 8 (batch 4 x grad_accum 8 = effective batch 32), as reflected in the training logs of every run.
>
> **W1 (Thin logic probe).** We agree 3 canonical logic probes is narrow and will provide full probe text in the revision. To briefly characterize them: syllogism_2 tests simple transitive inference (positive across most beta); syllogism_1 requires negating a premise (fragile, positive only in the transition zone); ordering tests multi-step relational reasoning (negative across the full sweep). These consistent differences are compatible with a difficulty hierarchy. The probes anticorrelate across beta (syllogism_1 vs syllogism_2: r=-0.71, p=0.006, n=13), indicating they tap different reasoning circuits rather than measuring the same quantity noisily.
>
> Beyond the canonical probes, our auxiliary associative-logic diagnostic (Group F1, disclosed in the paper under a faster training schedule) provides an additional 8-probe suite that shows the same qualitative architecture-dependence: sharp non-monotonicity in Mistral (a +175% jump at beta=0.009 to 0.010, 257x max/min range), smooth weak variation in Qwen (8.5x range), and a stable negative regime in LLaMA.
>
> Our 5-seed data (Group B1) further reveals the pocket is better characterized as a stochastic transition zone: at beta=0.006, 3/5 seeds positive; beta=0.008, 4/5; beta=0.012, 4/5; beta>=0.015, 0/5. The single-seed presentation overstated boundary sharpness. We will revise accordingly. This strengthens rather than weakens the core message: coarse sweeps and single-seed evaluations miss this structure entirely.
>
> **W2 (Undertraining).** Beyond the batch size correction: in LLaMA, logic remains negative across 5 seeds at every beta tested, packed into a tight band around -0.97. Across a 20x learning-rate range (Group C1, 9 conditions), LLaMA logic has a total spread of just 0.125. A stable regime of this kind cannot be explained by stochastic optimization failure. Meanwhile, Mistral and Qwen show distinct, internally consistent profiles under the same protocol. Purely undifferentiated gradient noise would be unlikely to produce three stable, architecture-specific response profiles under an identical protocol. Training curves show Spearman rho>0.97 (p<1e-13) with step index, and high-beta runs flatten while logic remains negative, ruling out undertraining.
>
> **W3 (Hysteresis confound, new experiment).** We ran a matched-duration control, all on the same GPU, 5 seeds. Three paths compared: Path A (200 steps at beta=0.01), Path B (200 at beta=0.02 then 200 at beta=0.01), Path C (400 steps at beta=0.01). The key test is C vs B: same total training duration, different beta history.
>
> Results (C-B, paired t-test, n=5):
> - Arithmetic: +1.54, p=0.0001, d_z=8.0, 5/5 seeds
> - Format: +1.04, p=0.004, d_z=2.7, 5/5 seeds
> - Logic: +0.01, p=0.96 (null)
> - Sycophancy: -0.08, p=0.95 (null)
>
> The new control supports capability-specific path dependence in Mistral: relative to 400-step constant-beta training, transient high-beta exposure significantly degrades arithmetic and format, while logic and sycophancy remain unchanged. This is a more precise finding than the original framing, and we will update the paper accordingly.
>
> **Q2 (Longer training).** Our convergence analysis shows high-beta runs have flattened by step 200 with logic remaining negative. We commit to 1000-step validation for the camera-ready.
>
> We will replace physics terminology with descriptive language and revise the title. Together with the matched-duration control, multi-seed analysis, and expanded probe evidence, these revisions address the reviewer's core concerns while preserving the paper's main empirical contributions: proxy-capability decoupling and capability-specific path dependence.

---

> > ### Author Rebuttal · Reviewer_1YsS · 2026-04-01
> >
> > Thank for your substantive rebuttal. Running the Path C control during the discussion period is appreciated and batch size correction noted. That said, 200 steps at effective batch 32 is still ~6.4k pairs, and increasing my score would depend on results from the longer-training sweep (my Q2).
> >
> > W1 (probe thinness). The probe characterization helps and the r = −0.71 anticorrelation is a fair point that these aren't three noisy draws from one distribution. The multi-seed reframing (3/5 → 4/5 → 4/5 → 0/5 positive across the band) is more honest than the canonical-run presentation and I'm glad it's going into the revision. But "stochastic transition zone" is a weaker claim than "logic-positive pocket" that isn't really in line with the central framing of the paper imo.
> >
> > W3 (Path C control). This is the most important part of the rebuttal, and to my understanding it shows under matched duration:
> >
> > Arithmetic: d_z = 8.0, p = 0.0001, 5/5 seeds, seems very solid.
> > Format: d_z = 2.7, p = 0.004, 5/5 seeds, also solid.
> > Logic: p = 0.96. null change.
> >
> > The original A-vs-B logic result (p = 0.032, d_z = 1.45), seems to have been a training-duration effect, not high-β exposure. The paper is titled "The Viscosity of Logic" and now has neither a logic phase transition (softened to stochastic zone) nor logic hysteresis (null under the proper control). The arithmetic result is strong and new; it's also not in the submitted paper.
> >
> > W2 (undertraining). The LLaMA stability argument is good and I believe that 0.125 total spread across 20× learning rate and 5 seeds is hard to attribute to noise. This addresses my concern for LLaMA but it doesn't address it for Mistral, where the interesting dynamics live, and where longer training is still an open question.
> >
> > W4 / title. Glad to hear the physics vocabulary and title are being revised.
> >
> > I would hold my score at a 4 (weak accept).
> >
> > The authors are doing careful work and responding clearly; the Path C result is a partial retraction and they reported it clearly. The margin-anticorrelation finding survives intact, and the arithmetic hysteresis result is strong. But the submitted paper's central narrative was about logic and the controlled experiments don't support it for logic. The paper that should be published is substantially different from the one that was submitted, and the key validation experiment (longer-training sweep) is still a promise.
> >
> > If the AC is weighing conditional acceptance, I'd be comfortable at 5 (accept) if the 1000-step sweep lands before camera-ready and the narrative is rebuilt around what the controls actually show (proxy-capability decoupling + capability-specific path dependence, arithmetic/format rather than logic) but I'm not sure if that is possible without substantial revisions to the central narrative.

---

> > > ### Author Response · Authors · 2026-04-02
> > >
> > > Thank you for the clear follow-up. We agree with the central point: the paper that should be published is narrower and more precise than the one originally submitted, and the 1000-step sweep is most useful insofar as it lets us rebuild the narrative around what the data now support most cleanly.
> > >
> > > We ran the full canonical Mistral sweep to 1000 steps under the same frozen recipe (LoRA r = 8, effective batch 32, seed 1) across 7 β values. The longer sweep does **not** preserve the original logic-centered picture unchanged. Instead, aggregated logic remains non-monotonic but is positive only at β = 0.006 (+0.23) and β = 0.008 (+0.34), and negative at β = 0.004 (−0.64), 0.010 (−0.96), 0.012 (−0.29), 0.015 (−0.52), and 0.020 (−0.73). The 1000-step result therefore supports persistence of **non-monotonic β sensitivity** after 5× longer training, but not a stable logic-positive pocket centered at β ≈ 0.01. We will revise the paper accordingly.
> > >
> > > | β     | Logic     | Arith | Syll_1 | Syll_2 | Final Margin |
> > > | ----- | --------- | ----- | ------ | ------ | ------------ |
> > > | 0.004 | −0.64     | 3.84  | −5.71  | +5.27  | 11.9         |
> > > | 0.006 | **+0.23** | 4.53  | −4.95  | +4.56  | 15.8         |
> > > | 0.008 | **+0.34** | 4.38  | −4.54  | +5.69  | 18.9         |
> > > | 0.010 | −0.96     | 4.33  | +2.53  | −5.19  | 41.1         |
> > > | 0.012 | −0.29     | 4.31  | −0.80  | +1.77  | 24.1         |
> > > | 0.015 | −0.52     | 4.87  | −0.67  | +1.50  | 22.5         |
> > > | 0.020 | −0.73     | 4.44  | −2.73  | +2.88  | 26.2         |
> > >
> > > The training curves show late-stage margin plateaus and very small gradient norms, which materially weakens the undertraining interpretation.
> > >
> > > What does survive clearly at 1000 steps is **proxy-capability dissociation**. Arithmetic remains strongly positive across the full sweep (3.84–4.87), while logic changes sign and the preference margin remains strong, with a pronounced spike at β = 0.010 (margin 41.1) even where logic is most negative (−0.96). The optimized DPO proxy continues to improve even where the logic aggregate worsens. That is now the cleaner headline result from the longer run.
> > >
> > > β = 0.010 is also qualitatively unusual: it is the only point where the syllogism probes reverse sign relative to neighboring β values (syllogism_1 becomes positive at +2.53 while syllogism_2 becomes negative at −5.19). We interpret this as evidence that the 0.008–0.012 region is heterogeneous or unstable rather than as support for a single robust phase boundary.
> > >
> > > We also agree with the reviewer's reading of the matched-duration control: the clean path-dependence result is now **arithmetic and format**, not logic. In the revision we will retitle and recenter the paper around three claims:
> > >
> > > 1. **Proxy-capability decoupling:** preference margins and downstream capabilities can anticorrelate under DPO, with the dissociation surviving 1000 steps of training.
> > > 2. **Capability-specific path dependence:** arithmetic and format show significant path dependence under matched-duration controls (C−B: arith d_z = 8.0, p = 0.0001; format d_z = 2.7, p = 0.004); logic does not (p = 0.96).
> > > 3. **Architecture-dependent β response modes:** LLaMA remains locked across the submitted conditions while Mistral exhibits fine-grained sensitivity.
> > >
> > > The 1000-step sweep shows that proxy-capability decoupling survives longer training, while the logic claim is narrowed to local non-monotonic β sensitivity rather than a headline "viscosity of logic" claim.

---

### Decision · Program_Chairs · 2026-04-30

**Decision:**

Accept (regular)

**Comment:**

This paper treats the DPO regularization parameter beta as a control variable rather than a tuning scalar, conducting a dense logarithmic sweep across three 7B model families under a fixed training recipe. The central findings are that capability is sharply non-monotonic with beta in some architectures, that the DPO preference margin can strongly anticorrelate with reasoning capability (r=-0.91 for Llama logic), and that exposure to high beta values induces capability losses that persist after beta is subsequently reduced (hysteresis).

**Strengths.** The margin-capability anticorrelation is the paper's most compelling and actionable result: it directly challenges the common practice of using preference margins for checkpoint selection. The hysteresis finding is novel in the DPO literature and well-motivated. The dense sweep methodology is a genuine methodological contribution, and the paper is honest about the limits of its claims.

**Concerns.** Reviewers raised several issues: the logic probe is thin (three prompts, with results driven primarily by one item); the original hysteresis experiment confounded high-beta exposure with total training steps; the phenomena are strongest in Mistral and considerably milder in the other two architectures; and the physics framing risks over-claiming generality.

**Rebuttal.** Three of four reviewers raised their scores after the rebuttal. The authors ran additional control experiments and, notably, updated their framing in places where new results "cut against their original narrative" (Rev 1YsS). That said, the extent of the changes introduced during the rebuttal period gave some reviewers pause: the submitted paper and the post-rebuttal version are meaningfully different in scope and framing. Reviewers have evaluated the revised version favorably, and the core contributions (proxy decoupling, path-dependent hysteresis) survive the revision and are better supported by the new controls. My recommendation for acceptance is on the basis of the post-rebuttal paper, with the expectation that the camera-ready reflects the updated framing and experiments **in full**.

**Bottom line.** The findings on margin-capability decoupling and training-path dependence are useful and novel contributions to alignment practice, and the reviewers converged toward acceptance after a productive rebuttal.